# Cross-Talk between NADPH Oxidase and Mitochondria: Role in ROS Signaling and Angiogenesis

**DOI:** 10.3390/cells9081849

**Published:** 2020-08-06

**Authors:** Tohru Fukai, Masuko Ushio-Fukai

**Affiliations:** 1Vascular Biology Center, Departments of Pharmacology and Toxicology, Medical College of Georgia at Augusta University, Augusta, GA 30912, USA; tfukai@augusta.edu; 2Charlie Norwood Veterans Affairs Medical Center, Augusta GA 30901, USA; 3Department of Medicine (Cardiology), Medical College of Georgia at Augusta University, Augusta, GA 30912, USA

**Keywords:** NADPH oxidase, mitochondria, reactive oxygen species, angiogenesis, redox signaling, endothelial cell, vascular endothelial growth factor

## Abstract

Angiogenesis, a new vessel formation from the pre-existing ones, is essential for embryonic development, wound repair and treatment of ischemic heart and limb diseases. However, dysregulated angiogenesis contributes to various pathologies such as diabetic retinopathy, atherosclerosis and cancer. Reactive oxygen species (ROS) derived from NADPH oxidase (NOX) as well as mitochondria play an important role in promoting the angiogenic switch from quiescent endothelial cells (ECs). However, how highly diffusible ROS produced from different sources and location can communicate with each other to regulate angiogenesis remains unclear. To detect a localized ROS signal in distinct subcellular compartments in real time in situ, compartment-specific genetically encoded redox-sensitive fluorescence biosensors have been developed. Recently, the intercellular communication, “cross-talk”, between ROS derived from NOX and mitochondria, termed “ROS-induced ROS release”, has been proposed as a mechanism for ROS amplification at distinct subcellular compartments, which are essential for activation of redox signaling. This “ROS-induced ROS release” may represent a feed-forward mechanism of localized ROS production to maintain sustained signaling, which can be targeted under pathological conditions with oxidative stress or enhanced to promote therapeutic angiogenesis. In this review, we summarize the recent knowledge regarding the role of the cross-talk between NOX and mitochondria organizing the sustained ROS signaling involved in VEGF signaling, neovascularization and tissue repair.

## 1. Introduction

Angiogenesis is the process of a new blood vessel (capillaries) formation from pre-existing vasculature. It is involved in physiological neovascularization such as embryonic development and wound healing and is required for vascular restoration in response to ischemia via delivering oxygen and nutrients in ischemic cardiovascular diseases. Excess and pathological angiogenesis contribute to cancer, ocular diseases such as diabetic retinopathies and atherosclerosis [1]. Thus, enhancing or inhibiting angiogenesis has great therapeutic significance. Especially, peripheral arterial disease (PAD), which is caused by atherosclerotic occlusion in the lower extremities, results in decreased blood flow and amputation of the foot or leg. Thus, promoting angiogenesis to restore limb perfusion is an important therapeutic approach for the treatment of PAD. Vascular endothelial growth factor (VEGF) is a key angiogenic growth factor that stimulates disruption of the endothelial cell (EC) junction, EC migration, proliferation and capillary tube formation mainly through VEGF receptor type2 (VEGFR2/Flk1) [2]. VEGF binding to VEGFR2 on the plasma membrane induces receptor dimerization and autophosphorylation, which is followed by receptor internalization (endocytosis) to early endosomal compartments where sustained VEGFR2 signaling is further activated to drive angiogenesis [3]. Other growth factors such as fibroblast growth factor (FGF), platelet-derived growth factor (PDGF) and angiopoietin-1 as well as hypoxia also promote angiogenesis in ECs.

Reactive oxygen species (ROS) (e.g., superoxide anion (O_2_^−^) and hydrogen peroxide (H_2_O_2_)) are toxic in excess. However, ROS at the physiological level function is essential for redox signaling involved in various biological responses including cell proliferation, migration, differentiation and gene expression [4,5]. Accumulating evidence suggests that angiogenesis can be also stimulated by ROS derived from ECs and other cell types such as vascular smooth muscle cells, myeloid cells such as neutrophils and macrophages. There are many sources of ROS including NADPH oxidases (NOX), the mitochondrial electron transport chain (ETC), xanthine oxidase, uncoupled endothelial nitric oxide synthase (eNOS), cytochrome P-450 oxygenase and cyclooxygenase. The vascular NOX isoforms (Nox1, Nox2, Nox4 and Nox5) differ in their activity and cell specificity in response to agonists, growth factors and hypoxia, and the type of ROS released after activation. In ECs, ROS derived from NOX (especially Nox2 and Nox4) in plasma membranes or intracellular cytosolic compartments [5,6,7,8,9] as well as mitochondria [10,11,12] play a key role in angiogenic response induced by growth factors such as VEGF [13]. However, the mechanisms by which highly diffusible ROS generated from different sources (enzymes) and locations can coordinate and communicate to promote a sustained growth-related angiogenic process are poorly understood.

During the past several years, the cross-talk between NOX and mitochondria, termed “ROS-induced ROS release”, has been proposed as a positive feed-forward mechanism of localized ROS production to organize redox signaling [14,15]. For example, it is shown that H_2_O_2_ activates Nox2 or p22phox to produce O_2_^−^ in fibroblasts and smooth muscle cells [16]; uncoupled eNOS produces O_2_^−^, instead of NO, to enhance mitochondrial ROS (mitoROS) production [17,18]; NOX-derived ROS increase mitochondrial ROS [13,19]; and mitochondrial ROS stimulate NOX activation [20]. This feed-forward ROS-induced ROS release mechanism can be targeted to inhibit pathological angiogenesis associated with oxidative stress or promote ROS-dependent physiological and therapeutic angiogenesis. This review will summarize the recent knowledge regarding the role of ROS-induced ROS organized by the cross-talk between ROS derived from NOX (plasma membrane and cytosol) and mitochondria in driving angiogenesis, in particular, focusing on VEGF signaling. Understanding these mechanisms should provide new insights into therapeutic strategies for various angiogenesis-dependent cardiovascular diseases and cancer.

## 2. Role of ROS Signaling in Angiogenesis

The endothelium lines the blood vessel lumen to mediate blood flow homeostasis and tissue metabolic demands by supplying nutrients and oxygen. Healthy ECs are in quiescence but new vessel formation (angiogenesis) occurs to revascularize tissues in response to angiogenic growth factors such as VEGF released by hypoxia, nutrient deprivation or tissue injury [1]. This organized and coordinated angiogenic phenotype switch from quiescent ECs is impaired in various diseases such as diabetes and aging with endothelial dysfunction, which results in tissue ischemia, leading to ischemic cardiovascular diseases including PAD. By contrast, dysregulated and excess angiogenesis stimulates cancer and diabetic and ocular retinopathy. In adults, ROS at the appropriate level are increased in response to growth factors (e.g.,VEGF), ischemia or wound injury, which function as signaling molecules to promote the angiogenic switch from quiescent ECs (Figure 1). Excess ROS contribute to pathological angiogenesis involved in cancer, atherosclerosis and pathological retinopathy.

ROS include free radicals such as O_2_^−^, hydroxy radicals (OH) and non-radicals such as H_2_O_2_. The O_2_^−^ generated by various ROS enzymes is rapidly scavenged by superoxide dismutases (SODs including cytosolic SOD1 (Cu/ZnSOD), mitochondrial SOD2 (MnSOD), SOD3 (extracellular SOD, ecSOD, EC-SOD)) to generate H_2_O_2_ (Figure 2). H_2_O_2_ is further converted to H_2_O by catalase, glutathione peroxidase (GPX) and peroxiredoxins (PRXs) (Figure 2). Oxidative stress (excess ROS) contributes to pathological angiogenesis, such as tumorigenesis, diabetic retinopathy and developing hypertension, heat failure and vasculopathy [21,22,23]. Further, abnormal angiogenesis induced by oxidative stress plays causative roles to promote atherosclerosis by increasing both macrophage infiltration and the thickening of the blood vessel wall by oxidized low-density lipoproteins (LDLs) [22,24,25,26].

By contrast, ROS, especially H_2_O_2_, at optimal levels function as signaling molecules to mediate various biological responses including angiogenesis [4,5], which is required for tissue repair and remodeling [27,28,29]. For example, exogenous ROS increase VEGF or VEGFR2 expression [30] and stimulate EC proliferation and migration [31,32]. During wound healing, ROS induced by tissue hypoxia induce angiogenesis by stimulating VEGF production from macrophages, fibroblasts, ECs and keratinocytes [21,33,34]. We reported that endothelium-derived H_2_O_2_ is required for post-ischemic neovascularization in vivo by using EC-specific catalase-overexpressing mice [35]. Furthermore, we reported that VEGF-induced ROS are required for VEGFR2 tyrosine phosphorylation, EC migration and proliferation via activation of small GTPase ARF6 localized at caveolae/lipid rafts in ECs [36]. Furthermore, it is shown that ROS-mediated redox signaling linked to angiogenesis involves mitogen-activated protein kinases (MAPKs), PI3 kinase, Akt, JAK-STAT, protein tyrosine phosphatases (PTPs) such as protein tyrosine phosphatase 1B (PTP1B) and SH2-containing protein tyrosine phosphatase 2 (SHP2), phosphatase and tensin homolog (PTEN) as well as transcription factors including HIF-1α, NFkB and AP-1.

To function as secondary messengers, H_2_O_2_ promotes signaling by oxidizing hyperactive cysteine residues on target proteins, which exist as thiolate anions (Cys-S-) at physiological pH. Of note, most cysteines have a pKa over 8, and therefore they remain protonated at physiological pH. ROS oxidize the thiolate anion to the sulfenic acid (Cys-OH), which forms a disulfide bond and alters redox signaling. Under normal conditions, this process is reversible by the disulfide reductases thioredoxin (TRX) and glutaredoxin (GRX). However, under pathological conditions with excess ROS (oxidative stress) and reduction of Trx and Grx, irreversible hyperoxidation of Cys residues (sulfinic and sulfonic acid) can be formed. Using “redox-dead” Cys17Ser PKARIα knock-in mutant mice, Burgoyne et al. [37] reported that PKARIα oxidation and dimerization-mediated activation is involved in VEGF-, tumor- and ischemia-induced angiogenesis. By contrast, the cytosolic receptor tyrosine kinase domain of VEGFR2 has two oxidation-sensitive Cys residues and is kept in a reduced state by antioxidant enzyme PRX-2 in quiescent ECs. However, Prx2 deficiency in quiescent ECs increases Cys oxidation of VEGFR2, thereby forming disulfide bonds, which in turn inactivates VEGFR2 that cannot respond to VEGF [38]. Thus, keeping quiescent ECs at a reduced state is important for driving an ROS-dependent VEGF-VEGFR2-stimulated angiogenic switch. Recently, we reported that redox-sensitive Cys residues of mitochondria fission protein Drp1 are kept in a reduced state by ER-localized thiol oxidoreductase protein disulfide isomerase (PDIA1) in quiescent ECs [39]. Reduction of PDIA1 in diabetes increases Cys-OH formation of Drp1 at the mitochondria-associated membrane (MAM), thereby promoting mitochondrial fragmentation and excess mitoROS production, which results in EC senescence and impaired angiogenesis [39]. These results suggest that redox regulation of the ER–mitochondria cross-talk at MAM by oxidoreductase PDI plays an important role in maintaining quiescent EC integrity. This might be required for driving efficient angiogenic responses induced by ROS-generating angiogenic growth factor VEGF.

## 3. Role of NOX in Angiogenesis

NOX generates O_2_^−^ or H_2_O_2_ by catalyzing the transfer of electrons from NADPH to reduce oxygen via their NOX catalytic subunit. The NOX family consists of seven isoforms including Nox1, Nox2, Nox3, Nox4, Nox5 and Duox1/Duox2 which localize at distinct subcellular compartments within the cells [40,41,42]. Their expression level is organ- and cell type-specific, and the type of ROS released and regulation of their activity are also different. The transmembrane catalytic subunits Nox1, Nox2, Nox3 and Nox4 interact with the small transmembrane regulatory subunit p22phox, while Nox5, which is not expressed in rodents, does not require a regulatory subunit for its activation. Duox1 and 2 require Duoxa1 and Duoxa2 as a scaffold for their function [41,43]. Each NOX contains six transmembrane domains and a cytoplasmic domain that binds NADPH and flavin adenine dinucleotide, and each isoform has specific cytosolic subunits.

In ECs, major source of ROS are Nox1, Nox2, Nox4 and Nox5 [41,42,43,44,45]. Nox4 is the most highly expressed NOX in ECs. Although Nox1, Nox2 and Nox5 are O_2_^−^-generating enzymes, Nox4 overexpression increases primarily H_2_O_2_ [40,41,42,46] due to an extended extracytosolic loop [47]. Nox5 is activated in a Ca^2+^-calmodulin-dependent manner [40,41,48] (Figure 3). Prototype Nox2 (gp91phox) was first discovered as a phagocytic oxidase to kill bacteria by generating O_2_^−^ [49] and consists of membrane-bound catalytic subunit Nox2 and regulatory small subunit p22phox, and cytosolic subunits p47phox, p67phox and a small G protein Rac to produce O_2_^‒^ that is rapidly dismutated by SODs to generate H_2_O_2_ [50] (Figure 1 and Figure 2). Nox1 and Nox4 catalytic subunits also couple with p22^phox^, but Nox1 oxidase activity requires cytosolic NOX activator 1 (NOXA1) and NOX organizer 1 (NOXO1), which are isoforms of p47^phox^ and p67^phox^, respectively (Figure 3). Nox4 is constitutively active without classical cytosolic subunits but is activated by polymerase delta-interactive protein 2 (Poldip2) (Figure 3). Poldip2 was originally reported as a cytosolic binding partner for the Nox4-p22phox complex in vascular smooth muscle cells (VSMCs) [51]. However, recent studies show that Poldip2 is a mitochondrial protein that regulates the activity of the TCA cycle and metabolic reprograming [52] as well as a regulator of the differentiated phenotype in VSMCs [53]. Using Poldip2-depleted HUVECs and Poldip2^+/−^ mice, Poldip2 is shown to be involved in serum-induced EC proliferation and post-ischemic neovascularization [54]. Another Nox4-interacting protein is calnexin, which may be needed for the proper maturation, processing and function of NOX4 in the endoplasmic reticulum (ER) [55] (Figure 3).

NOX-derived ROS are required for angiogenic response induced by various growth factors (e.g., VEGF, angiopoietin, PDGF, TGFβ, etc.) and hyperoxia [13,56,57,58,59,60,61,62,63]. Nox4^−/−^ or Nox2^−/−^ mice, EC-specific Nox4, dominant-negative Nox4 or EC-specific catalase-overexpressing mice [35,41,64,65] reveal that Nox2 or Nox4 or their regulators are required for ROS-dependent angiogenic signaling in ECs, tumor angiogenesis as well as post-ischemic neovascularization using a PAD model [8,36,59,62,65,66,67,68,69,70,71,72,73,74,75]. However, Nox2 and Nox4 also induce EC dysfunction depending on their subcellular localization, extent and duration of activation [45,76]. Indeed, hyper Nox2 activation which produces overproduction of ROS contributes to various pathologies such as diabetes, hypertension and ischemic stroke [77,78,79,80,81,82].

Compartmentalization of the ROS signal is essential for specific activation of redox signaling after receptor activation. Nox2 and Nox4 exist in diverse subcellular compartments such as plasma membranes including caveolae/lipid rafts, endosomes, ER and mitochondria [6,7,8,9,83,84]. During directional migration, Nox2 translocates to lamellipodia and membrane ruffles via binding to p47phox with the scaffold proteins TNF receptor associated factor 4 (TRAF4) and WASP family verprolin homologous protein 1 (WAVE1)/Rac1, a cytosolic component of Nox2. Using a cell-permeable biotin-labeled Cys-OH trapping probe [85], we showed that VEGF stimulation increases Cys-OH formation of scaffold protein IQGAP1 that binds to active VEGFR2 and Rac1 at the lamellipodial leading edge, which promotes directional EC migration [66,86,87]. We also found that IQGAP1-deficient mice show impaired post-ischemic neovascularization using a PAD model [88]. These results suggest that Nox2 binding to adaptor/scaffold proteins which translocate to the lamellipodial leading edge is required for localized ROS production and Cys oxidation of ROS targets, thereby promoting EC migration and angiogenesis [6,7,8]. In addition, ROS induce oxidative inactivation of PTPs which are localized at distinct subcellular compartments. This establishes compartmentalization of ROS-dependent tyrosine kinase signaling pathways involved in angiogenesis.

ROS production is also localized via NOX interaction with signaling platforms associated with lipid rafts and caveolae as well as endosomes [6,8]. We demonstrated that ecSOD-derived H_2_O_2_ induces Cys oxidation/inactivation of PTP1B and density-enhanced phosphatase 1 (DEP1) in caveolae/lipid rafts, thus enhancing VEGFR2 signaling and angiogenesis in ECs, which is required for restoring neovascularization induced by tissue ischemia [89]. There is also evidence that Nox4 is found in the nucleus, indicating its involvement in redox-responsive gene expression [83]. Thus, targeting NOX or its binding partners to discrete subcellular compartments is a mechanism of localizing ROS production and its downstream redox signaling events involved in angiogenesis and other biological responses.

## 4. Role of Mitochondria-Derived ROS in Angiogenesis

Not only NOX but also mitochondria-derived ROS function as a tightly regulated redox signal that transmits information from the organelle to the cell. Mitochondria are redox-active organelles and transfer more than 90% of the electron to O_2_ to generate O_2_^−^ as the terminal electron acceptor [90]. The mitochondrial inner membrane contains five multiprotein complexes such as Complex I (NADH-quinone oxidoreductase), Complex II (succinate dehydrogenase), which transfers electrons into the chain from succinate, Complex III (coenzyme Q: cytochrome C oxidoreductase), Complex IV (cytochrome C oxidase) and Complex V (ATP synthase). During cellular respiration, the electrons released from the electron transport chain (ETC) react with O_2_ to produce O_2_^−^ [91]. Complexes I and III are the main sites of electron transfer to O_2_ to produce O_2_^−^ which is released into the intermembrane space (IMS) or matrix and is involved in redox signaling [92], because it has easier access to the cytosol. Since O_2_^−^ is a charged species, it cannot diffuse across mitochondrial membranes. Thus, the voltage-dependent mitochondrial anion channel (VDAC) seems to help in releasing the intermembrane mitochondrial O_2_^−^ to the cytosol [93]. In addition, O_2_^−^ produced in the mitochondrial matrix or IMS is rapidly converted to H_2_O_2_ by SOD2, or SOD1, respectively, and this H_2_O_2_ can diffuse through both inner and outer mitochondrial membranes to the cytosol to activate redox signaling. H_2_O_2_ is also further converted to H_2_O by GPX or PRX or catalase.

Mitochondria function as an O_2_ sensor and transmit a hypoxic signal by releasing ROS to the cytosol [94]. Hypoxia stimulates mitoROS production such as H_2_O_2_ from mitochondrial complex III and the ROS trigger HIF1-α stabilization [94,95,96], which in turn increases the transcription of angiogenic genes such as VEGF [97]. Thus, mitochondria regulate angiogenic responses by controlling cellular metabolism linked to mitoROS. In ECs, not only Nox2 and Nox4 but also mitoROS play an important role in VEGF- and angiopoietin 1-induced angiogenic responses [13,98,99]. In addition, mitochondrial respiratory chain complex III (QPC) is shown to be necessary for EC proliferation involved in retinal and tumor angiogenesis [100]. MitoROS also promote ligand-independent H_2_O_2_-induced transactivation of VEGFR2 [101]. By contrast, mitoROS generated in hyperglycemia induce ligand-independent but Src-dependent phosphorylation of VEGFR2, which reduces the amount of VEGER2 at the cell surface required for VEGF binding, thereby attenuating VEGF-induced pro-angiogenic effects in diabetes with oxidative stress [102]. These results suggest that mitoROS produced in physiological and pathological conditions have an opposite impact on VEGFR2 signaling and angiogenesis.

## 5. ROS-Induced ROS Release

It remains unknown how ROS derived from distinct compartments communicate and affect cell function differentially [22]. Zorov et al. reported that the photodynamically produced initial phase of mitoROS caused the mitochondrial permeability transition (MPT) with a delayed amplified phase of mitoROS generation in cardiac myocytes (termed mitochondrial “ROS-induced ROS release”) [14]. Chung et al. showed that sodium salicylate-induced ROS stimulate ROS as well as mitochondrial membrane potential collapse, which leads to cytochrome *c* release and caspase activation [103]. In cardiac cells, mitochondrial permeability transition pore (mPTP)-dependent and independent ROS-induced ROS are observed: Increased ROS induce mPTP-dependent mitochondrial depolarization, resulting in a short-lived ROS production derived from mitochondrial ETC. By contrast, mPTP-independent ROS stimulate the opening of the inner mitochondrial membrane anion channel, which in turn stimulates ETC-derived ROS release to the cytosol. This ROS-induced ROS release creates a positive feedback mechanism for enhanced ROS production among neighboring mitochondria, leading to mitochondrial and cellular injury [104]. The organelle excitability function for electrical and Ca^2+^ signals of mitochondria further amplify ROS signaling [105,106]. It is proposed that H_2_O_2_ induces cysteine oxidation of mitochondrial proteins including ETC proteins to enhance mitochondrial O_2_^−^ production [107], which is rapidly converted to H_2_O_2_ by SOD1 at IMS or SOD2 at the mitochondrial matrix, thereby further increasing mitochondrial ROS production. Thus, ROS-induced ROS release amplifies the ROS signal among each subcellular compartment [13,14,15].

There are several reports showing the role of ROS-induced ROS release involved in cardiovascular diseases (CAD). Li et al. [16]. demonstrate that exogenous exposure of VSMCs and fibroblasts to H_2_O_2_ induces O_2_^−^ production via non-phagocytic oxidase Nox2, thereby amplifying the vascular injury process. Zinkevich et al. reported that flow-induced H_2_O_2_ production and dilation in microvessels from CAD patients involves Nox2-derived ROS-induced mitochondrial ROS release [23]. In addition, Angiotensin II (Ang II)-induced Nox1 activation stimulates mitochondrial ROS, resulting in mitochondrial dysfunction and vascular senescence [108,109]. It is also reported that Nox2 stimulates mitoROS by activating reverse electron transfer and phosphorylation of cSrc in human aortic ECs, which contributes to Ang II-induced hypertension [110]. On the other hand, it is shown that mitochondrial ROS can stimulate NOX-derived ROS. For example, mitoROS induce activation of Nox in phagocytes and cardiovascular tissues, which in turn results in immune cell activation and development of Ang II-induced hypertension [20,111,112]. Thus, cross-talks between mitochondria and NOX in pathological conditions may represent a feed-forward vicious cycle to amplify excess ROS leading to oxidative stress, which can be a therapeutic target [19] (Table 1).

## 6. The Crosstalk between NOX and Mitochondria (ROS-Induced ROS Release) in Angiogenesis

The role of ROS-induced ROS release in VEGF signaling and angiogenesis remains elusive. To address this question, we performed real-time imaging by using cytosol- and mitochondria-targeted ratiometric redox-sensitive green fluorescent proteins (RoGFP) biosensors in human ECs [113]. This method allows us to determine the temporal–spatial relationship for VEGF-induced ROS production from the different subcellular compartments. We found that VEGF stimulation in human ECs rapidly increases NOX-derived H_2_O_2_ in the cytosol as shown by the PEG-catalase-inhibitable cytosolic RoGFP oxidation (first phase), followed by sustained mitoH_2_O_2_ production as shown by the mitochondrial RoGFP oxidation [13] (second phase) (Figure 4). With other data using gain and loss of function approaches for Nox4 or Nox2, we demonstrated that Nox4-derived H_2_O_2_ stimulates Nox2 to increase mitoROS, which promotes sustained VEGFR2-mediated angiogenic responses in ECs [13]. An alternative method to monitor intracellular H_2_O_2_ in different compartments includes a redox-active biosensor, Hyper, but the signal can be affected by pH changes and is insensitive to reducing stimuli. Recently, an improved version of Hypers has been reported to overcome these problems [114,115,116]. Consistent with our report, Evangelista et al. [65] also reported that Nox4-derived H_2_O_2_ can activate Nox2, which contributes to VEGF-induced S-glutathiolation of the sarco(endo)plasmic reticulum Ca^2+^-ATPase (SERCA) and EC migration likely at MAM [117,118]. Thus, these finding suggest that ROS-induced ROS release orchestrated by Nox4, Nox2 and mitochondria plays an important role in driving angiogenic phenotypes from quiescent ECs (Table 1).

It remains unclear how VEGF rapidly activates Nox4 to induce H_2_O_2_ in ECs. It is reported that insulin-like growth factor 1 (IGF-I) induces rapid Nox4 Tyr^491^ phosphorylation, which promotes rapid and localized ROS production via Nox4 binding to the adaptor protein growth factor receptor bound protein 2 (Grb2), which is in a multifunctional transmembrane glycoprotein, SH2 domain-containing protein tyrosine phosphatase substrate 1 (SHPS-1) complex. Thus, similar mechanisms may be involved in VEGF-induced rapid Nox4-mediated H_2_O_2_ production in ECs. It also remains unclear how Nox4-derived H_2_O_2_ can activate Nox2. Based on the literature, it is possible that H_2_O_2_ produced by Nox4 activation may activate Nox2 via phosphorylation of Nox2 or its cytosolic organizers such as p47phox and Rac1 [41,98,119]. To support this, it is shown that H_2_O_2_-induced activation of redox-sensitive kinase cSrc can phosphorylate p47phox [41] and Rac1 guanine nucleotide exchange factor Vav2 [120] that activates Rac1 in ECs [121], which in turn leads to activation of NOX2.

## 7. The Role of p66Shc in Crosstalk between NOX and Mitochondria Promoting Angiogenesis

The mechanism by which Nox4/Nox2-derived ROS stimulate mitoROS production remains unclear. In addition to ETC, one of key regulators of mitoROS is an adaptor protein, p66Shc. Once p66shc is phosphorylated at the Serine (Ser) 36 residue in the cytosol, it translocates to mitochondria where it oxidizes cytochrome c to generate H_2_O_2_ (Figure 5) [122,123].

We reported that VEGF induces rapid Rac1 activation by interaction with the non-phosphorylated form of p66shc, leading to Nox2-dependent ROS production, which contributes to VEGFR2 phosphorylation at caveolae/lipid rafts and subsequent angiogenic responses in ECs [124]. It is also shown that p66shc regulates cytosolic Nox organizer p47phox expression, which in turn regulates ROS generation [22,125]. We also reported that VEGF increases phosphorylation of p66Shc at Ser36 [124] by protein kinase C (PKC) and extracellular signal regulated kinase (ERK)/Jun *N*-terminal kinase (JNK), which are activated by Nox-derived H_2_O. This in turn increases mitoROS production that promotes sustained ROS-dependent VEGFR2 signaling and angiogenic responses [13]. This may represent a novel feed-forward mechanism of ROS-induced ROS release mediated through NOX–mitochondria cross-talk orchestrated by p-p66shc, which drives sustained growth-related angiogenic signaling programs in ECs [13,113] (Figure 6).

Furthermore, using binary (Tet-ON/OFF) conditional EC-specific Nox2 transgenic mice, Shafique et al. reported that the duration of the increase in NOX2-derived ROS determines the level of mitoROS and their paradoxical effects (beneficial vs. harmful) on the coronary endothelium [126]. They showed that short-term (eight weeks) increases in NOX2-ROS induce the AMP-activated protein kinase (AMPK)-eNOS-NO axis with low mitoROS, thereby promoting angiogenesis. However, long-term (20 weeks) increases in cytosolic NOX2-ROS result in nitro-Tyrosine-mediated inactivation of MnSOD, leading to a sustained mitoROS increase that induces the loss of mitochondrial membrane potential, which in turn inhibits angiogenesis. These results suggest that the Nox2-derived ROS duration can regulate mitoROS levels and their fate in ECs. However, it remains unknown whether short-term and long-term increased NOX2-ROS mice models may reflect endogenous physiological and pathological (i.e., diabetes or aging) conditions, respectively, or levels of p-p66Shc at different extents or durations.

## 8. Summary and Future Perspectives

From this review, it is clear that ROS-induced ROS release by the cross-talk between NOX- and mitochondria-derived ROS is essential for sustained angiogenic signaling, reparative angiogenesis and homeostatic maintenance of healthy vasculature. On the other hand, it can be toxic by amplifying ROS in excess, which will ultimately contribute to pathological angiogenesis and tissue damages in CAD, diabetes and aging. Multifaced ROS targets such as sulfenylated proteins or S-glutathionylated proteins or other cysteine-oxidized proteins in ROS-induced physiological or pathological angiogenesis are still poorly understood. Therefore, the systemic approach and measurement of ROS using compartment-specific redox-sensitive fluorescence biosensors in real time may need to understand the temporal and spatial modulation of ROS production and signaling involved in angiogenesis and vascular repair. In addition, using ROS or ROS-generating/blocking agents for therapy does not work always efficiently due to the lack of technology or information for their delivery to the distinct subcellular compartments. Thus, developing a new therapeutic strategy which specifically targets the ROS-induced ROS release mechanisms using specific inhibitors of NOX or mitochondria is important for treatment of various angiogenesis-dependent diseases such as ischemic heart and limb diseases, diabetic retinopathy and cancer.

## Figures and Tables

**Figure 1 cells-09-01849-f001:**
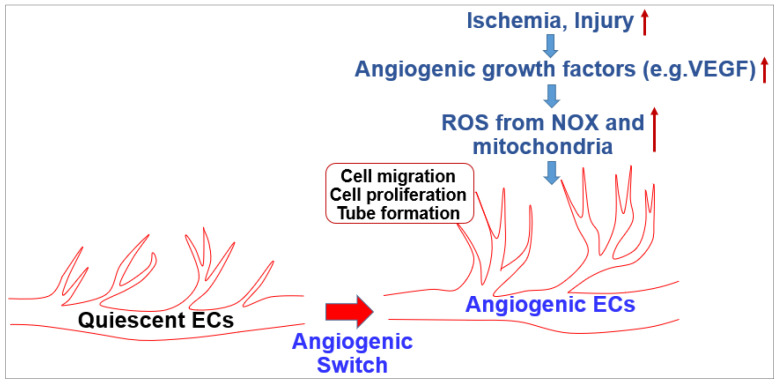
Role of reactive oxygen species (ROS) in angiogenesis in endothelial cells (ECs).

**Figure 2 cells-09-01849-f002:**
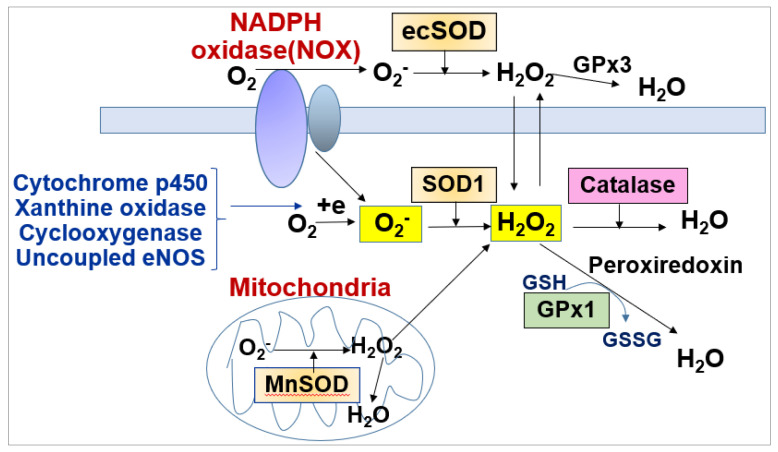
Generation and metabolism of ROS in ECs.

**Figure 3 cells-09-01849-f003:**
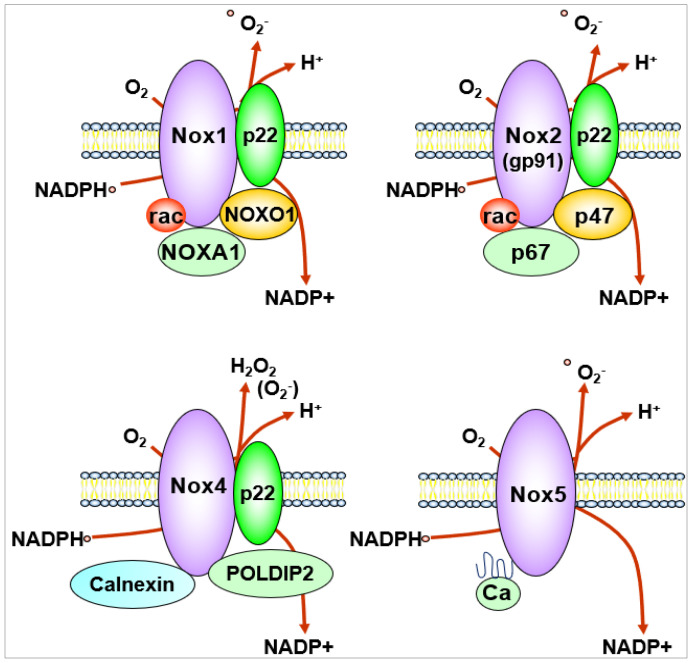
NOX isoforms expressed in endothelial cells.

**Figure 4 cells-09-01849-f004:**
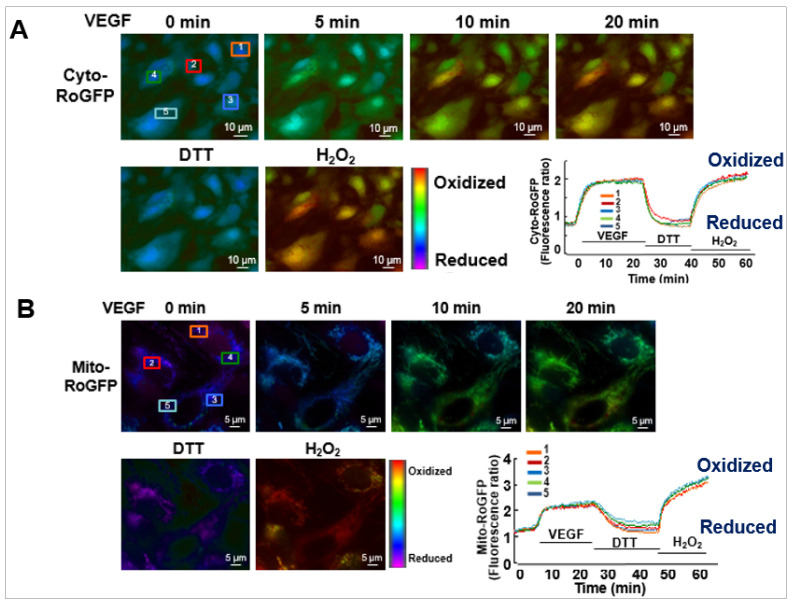
Real-time imaging for cytosolic and mitochondrial redox status in single cell in response to VGEF.

**Figure 5 cells-09-01849-f005:**
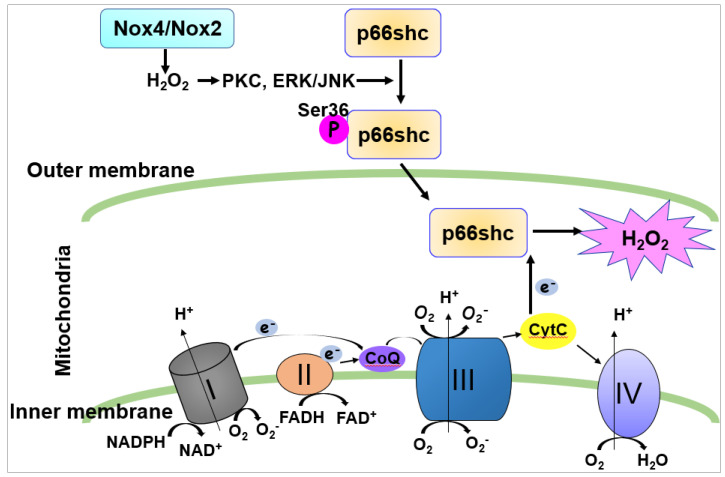
Mitochondrial H_2_O_2_ production via the Nox4/Nox2/p-p66Shc axis.

**Figure 6 cells-09-01849-f006:**
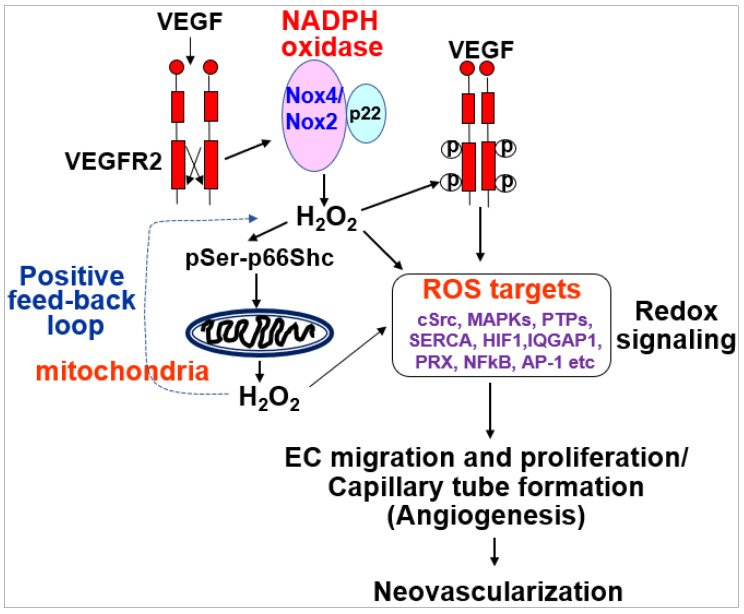
NOX–mitochondria cross-talk in VGEF signaling and angiogenesis.

**Table 1 cells-09-01849-t001:** Role of ROS-induced ROS release in angiogenesis and vascular disease.

	Cell Type	ROS-Induced ROS Release	Response or Function	Reference
1	VSMC/fibroblast	H2O2-p22phox(NOX2)-ROS	Cell injury and damage	[16]
2	adipose arterioles	Nox2-mitoROS	Flow-induced dilation	[23]
3	VSMCs	Nox1-mitoROS	AngII-induced senescence	[108]
4	leukocytes or aorta	p47phox (NOX2)-mitoROS	AngII-induced hypertension	[111]
5	VSMC	MitoROS-Nox1-ROS	Ang II-induced NOX activation	[112]
6	EC	Nox2-mitoROS	AngII-induced hypertension	[110]
7	EC	Nox4-Nox2-ROS	EC migration (angiogenesis)	[65]
8	EC	Rac1 (NOX2)-mitoROS	EC migration (angiogenesis)	[98]
9	EC	Nox4-Nox2-mtROS	Angiogenesis	[13]

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
