# Peer review of "Cross-Talk between NADPH Oxidase and Mitochondria: Role in ROS Signaling and Angiogenesis"

_cells, 2020, doi:10.3390/cells9081849_

Round 1

Reviewer 1 Report

In this review the authors “ summarize the recent knowledge regarding the role of cross-talk between NOX and mitochondria organizing the sustained ROS signaling involved in neovascularization and tissue repair”.

The authors made a good job in reviewing the literature.

In the opinion of this reviewer the generation of ROS by NOX and the cross-talk with mitochondria is abundantly and well described. Their role in angiogenesis, however, in not so clear  and some paragraphs are confusing for the non-specialized reader who wants to learn about the subject.

The authors may simplify the paragraphs that refer to angiogenesis and if they add a scheme about the effect of ROS on angiogenesis. it  would be very helpful. For instance the square box in Figure 5 could identify some proteins or transcription factors instead of stating  "ROS targets and redox signaling"

Other comments

1) line 105 it says:  Cysteine residues exist as a thiolate anion (Cys-S-) at physiological pH…. This sentence may be misleading. Most cysteines have a pKa over 8, and therefore they remain protonated at physiological pH.  Only some cysteines, by effect of surrounding groups, exhibit a lower pK a and exists as thiolate anion at physiological pH. These are the so called hyperreactive cysteines.

2) What do the authors mean by Ca2+ channel SERCA (line 294)

3) Some acronyms should be defined the first time they are mention

(PTP1B,  DEP1, MAM,  etc)

Some Cosmetic changes

Lines 104-115 and lines 118-131 are centered instead of justified.

Line 235  it reads Zorove; it should be Zorov

Line 248-252 there is a change in font

Author Response

The authors made a good job in reviewing the literature. In the opinion of this reviewer the generation of ROS by NOX and the cross-talk with mitochondria is abundantly and well described. Their role in angiogenesis, however, in not so clear and some paragraphs are confusing for the non-specialized reader who wants to learn about the subject.--The authors may simplify the paragraphs that refer to angiogenesis and if they add a scheme about the effect of ROS on angiogenesis. it would be very helpful. For instance, the square box in Figure 5 could identify some proteins or transcription factors instead of stating "ROS targets and redox signaling"

Response: According to the reviewer’s suggestion, we have revised the following issues.

  • We have revised the paragraph that refers to angiogenesis (page 3, first paragraph; page4, third paragraph; page 5, first paragraph) and added new scheme Figure 1 showing the role of ROS in angiogenesis in ECs for the non-specialized reader on this subject in the revised manuscript.

  • Subject title “6. p-p66Shc senses NOX-derived H2O2 signal to increase Mitochondrial ROS” was not clear to general reader. Thus, this title has been changed to “The role of p66Shc in crosstalk between NOX and mitochondria promoting angiogenesis” (page 15) in the revised manuscript.

  • We have added several ROS target proteins and transcription factors in revised Figure 6 (original Figure 5) in the revised manuscript.

Other comments

1) line 105 it says: “Cysteine residues exist as a thiolate anion (Cys-S-) at physiological pH…. This sentence may be misleading. Most cysteines have a pKa over 8, and therefore they remain protonated at physiological pH.  Only some cysteines, by effect of surrounding groups, exhibit a lower pK a and exists as thiolate anion at physiological pH. These are the so called hyperreactive cysteines.”

Response: Thank you for the suggestion. In the revised revision, we have corrected this issue (page 7, first paragraph) in the revised manuscript.

2) What do the authors mean by Ca2+ channel SERCA (line 294)

Response: We have changed to “sarco(endo)plasmic reticulum Ca2+-ATPase (SERCA)” in the revised manuscript (page 15, first paragraph).

3) Some acronyms should be defined the first time they are mention (PTP1B,  DEP1, MAM,  etc)

 Response: We have defined the all the acronyms in the revised manuscript.

Some Cosmetic changes

Lines 104-115 and lines 118-131 are centered instead of justified.

Response: We have justified them.

Line 235  it reads Zorove; it should be Zorov

Response: We have corrected this error.

Line 248-252 there is a change in font

Response: We have corrected this error.

Reviewer 2 Report

The authors paid special attention to cross-talk between reactive oxygen species (ROS) derived from NADPH oxidase (NOX) and mitochondria in neovascularization (angiogenesis) and tissue repair. Angiogenesis plays crucial roles for various physiological and pathological conditions, and its activities are regulated by complex mechanisms. As one of important regulators of angiogenesis, oxidative stress is suggested. In addition, mitochondria function is also known to be associated with biological activities of endothelial cells. Therefore, I think that their review has important information to understand the biological characteristics and treatment strategies of angiogenesis-related disease. However, I have a question before publication.

(Minor)

  1. I recommend adding the summary (Table) about the pathological roles of the cross-talk among NOS, ROS, and mitochondria in each angiogenesis-depended diseases.

Author Response

The authors paid special attention to cross-talk between reactive oxygen species (ROS) derived from NADPH oxidase (NOX) and mitochondria in neovascularization (angiogenesis) and tissue repair. Angiogenesis plays crucial roles for various physiological and pathological conditions, and its activities are regulated by complex mechanisms. As one of important regulators of angiogenesis, oxidative stress is suggested. In addition, mitochondria function is also known to be associated with biological activities of endothelial cells. Therefore, I think that their review has important information to understand the biological characteristics and treatment strategies of angiogenesis-related disease. However, I have a question before publication.

(Minor)

  1. “I recommend adding the summary (Table) about the pathological roles of the cross-talk among NOS, ROS, and mitochondria in each angiogenesis-depended diseases.”

Response: Thank you for the suggestion. We have included Table 1 summarizing “ROS-induced ROS release in angiogenesis and vascular disease” (page 13) in the revised manuscript.

Round 2

Reviewer 1 Report

All  comments raised in the  previous rewiew have been addressed.